# Cell Internalization in Fluidic Culture Conditions Is Improved When Microparticles Are Specifically Targeted to the Human Epidermal Growth Factor Receptor 2 (HER2)

**DOI:** 10.3390/pharmaceutics11040177

**Published:** 2019-04-11

**Authors:** Inmaculada Mora-Espí, Elena Ibáñez, Jorge Soriano, Carme Nogués, Thorarinn Gudjonsson, Leonardo Barrios

**Affiliations:** 1Unitat de Biologia Cel·lular, Departament de Biologia Cel·lular, Fisiologia i Immunologia, Facultat de Biociències, Universitat Autònoma de Barcelona, Bellaterra, 08193 Barcelona, Spain; xapaxin@gmail.com (I.M.-E.); Elena.ibanez@uab.cat (E.I.); jorge.soriano@uab.cat (J.S.); carme.nogues@uab.cat (C.N.); 2Biomedical Center, University of Iceland, 101 Reykjavík, Iceland; tgudjons@hi.is; 3Department of Anatomy, Faculty of Medicine, and Department of Laboratory Hematology, University Hospital, 101 Reykjavik, Iceland

**Keywords:** microfluidics, coculture, HER2, polystyrene µPs, biofunctionalization

## Abstract

Purpose: To determine if the specific targeting of microparticles improves their internalization by cells under fluidic conditions. Methods: Two isogenic breast epithelial cell lines, one overexpressing the Human Epidermal Growth Factor Receptor 2 (HER2) oncogene (D492HER2) and highly tumorigenic and the other expressing HER2 at much lower levels and non-tumorigenic (D492), were cultured in the presence of polystyrene microparticles of 1 µm in diameter, biofunctionalized with either a specific anti-HER2 antibody or a non-specific secondary antibody. Mono- and cocultures of both cell lines in static and fluidic conditions were performed, and the cells with internalized microparticles were scored. Results: Globally, the D492 cell line showed a higher endocytic capacity than the D492HER2 cell line. Microparticles that were functionalized with the anti-HER2 antibody were internalized by a higher percentage of cells than microparticles functionalized with the non-specific secondary antibody. Although internalization was reduced in fluidic culture conditions in comparison with static conditions, the increase in the internalization of microparticles biofunctionalized with the anti-HER2 antibody was higher for the cell line overexpressing HER2. Conclusion: The biofunctionalization of microparticles with a specific targeting molecule remarkably increases their internalization by cells in fluidic culture conditions (simulating the blood stream). This result emphasizes the importance of targeting for future in vivo delivery of drugs and bioactive molecules through microparticles.

## 1. Introduction

Drug targeting has the potential to improve the therapeutic efficacy and mitigate the non-specific effects of many drugs. In the last years, several types of drug delivery vehicles have been developed, including monoclonal antibodies [1], peptides [2], proteins [3], lipoproteins [4], carbohydrates [5], and polymeric nanoparticles [6,7]. Compared with the number of studies in which nanoparticles (NPs) are used [8,9,10,11,12], only a small number of studies involve the use of microparticles (µPs) [13,14,15]. It has been reported that small sizes and positive charges favor NP intake by cells [14,15,16], but in some cases, larger particle sizes could be advantageous for preventing non-specific interactions and internalization into normal non-phagocytic cells or for optimal tissue entrapment and transient retention [17]. Moreover, to target cancer cells, NPs and µPs surfaces can be modified to increase the interaction with plasma membrane-specific markers like the transferrin receptor [18], the folate receptor [19], or the human epidermal growth factor receptor 2 (HER2, also known as ERBB2) [20,21]. HER2 is a receptor tyrosine kinase which is overexpressed by some types of cancer cells and is considered a marker of poor clinical outcome in breast and ovarian cancer [22,23]. Some treatments directed to this target have already been approved and are clinically used, such as the anti-HER2 monoclonal antibody trastuzumab, alone or in combination with emtansine (T-DM1) [24], and the HER2 tyrosine kinase activity inhibitor lapatinib.

Traditionally, in vitro studies on drug carriers and drug release have been performed in static monolayer cell cultures. However, studies in microfluidic environments, mimicking the circulatory system, are currently gaining interest [12]. Compared with static cultures, microfluidic studies allow for better predictions about how a drug or a drug carrier running in a circulating flow, will interact with cells [25,26,27]. On the other hand, because normal and tumoral cells are intermingled in vivo, cocultures of normal and tumoral cells can better simulate tissue conditions than monocultures [28,29].

The overall objective of the present study was to evaluate the efficiency of targeting µPs to cells in physiological-like conditions (fluidic culture conditions). Two isogenic breast epithelial cell lines were used, one normal (D492) and the other overexpressing HER2 (D492HER2). Moreover, polystyrene µPs of 1 µm in diameter were biofunctionalized with a specific targeting protein, an anti-HER2 antibody, or with a non-specific secondary antibody. The specific objectives of the study were to evaluate and compare the cell internalization of these µPs in different culture conditions: monoculture versus coculture conditions, and static versus fluidic culture conditions.

## 2. Material and Methods

### 2.1. Biofunctionalization of Polystyrene µPs

Carboxylate polystyrene µPs of 1 µm in diameter (Polybead^®^ Carboxylate Microspheres. Polysciences, Inc., Warrington, PA, USA) were biofunctionalized with two different targeting molecules: (1) mouse anti-c-ERBB2/c-Neu (Ab-5) clone TA-1 (Millipore, Darmstadt, Germany), herein referred to as antiH, and (2) goat anti-mouse IgG2a secondary antibody Alexa Fluor^®^ 647 conjugate (Life Technologies, Carlsbad, CA, USA), herein referred to as secAb. Biofunctionalization was carried out using the PolyLink Protein Coupling Kit for COOH Microspheres (Polysciences) according to the manufacturer’s instructions. The size of the µPs before and after biofunctionalization was analyzed by transmission electronic microscopy (TEM) (JEOL, JEM 2011). Biofunctionalization was evaluated under a fluorescence inverted microscope (Olympus IX71, Olympus, Hamburg, Germany) and by the change in the ζ-potential.

Microscopically, biofunctionalization of µPs with secAb (µP-secAb) was evaluated directly on the basis of their far-red fluorescence emission. On the other hand, µPs biofunctionalized with antiH (µP-antiH) were incubated for 5 min with chicken anti-mouse IgG (H+L) secondary antibody Alexa Fluor^®^ 488 conjugate (1:500. Life Technologies) before the evaluation of green fluorescence emission.

Biofunctionalized and non-biofunctionalized µPs were separately resuspended in H14 culture medium [30] and sonicated for 5 min (Fisherbrand FB15047, Fisher Scientific, Germany) to achieve a monodispersed sample. Their ζ-potential was then measured with a Zetasizer Nano ZS (Malvern Instruments, Malvern, UK).

### 2.2. Cell Lines

Two isogenic breast epithelial cell lines, D492 and D492HER2, were used in the study. D492 is a non-tumorigenic cell line with stem cell properties that expresses low levels of the HER2 oncogene [30,31]. D492HER2 was generated by overexpressing the HER2 oncogene in D492 and is highly tumorigenic [32]. Both cell lines constitutively express green fluorescent protein (GFP).

The cells were cultured in serum-free H14 culture medium [30,31] at 37 °C and 5% CO_2_ (standard conditions). As explained below, cell culture was performed as follows: mono- or cocultures in static conditions, and cocultures in fluidic conditions.

### 2.3. Cell Cultures in Static Conditions

For monoculture experiments, cells were seeded at a density of 60,000 cells/well in 24-well plates (µ-Plate 24-Well ibiTreat: #1.5 polymer coverslip. ibidi, Martinsried, Germany). For coculture experiments, 30,000 cells of each cell line (D492 and D492HER2) were seeded together in each well. In both cases, the cells were maintained for 24 h in standard culture conditions prior to performing any experiments.

To analyze µP internalization, µPs (µP-antiH or µP-secAb) were sonicated for 5 min, diluted (1:100), counted with a hemocytometer, and then added at a proportion of 45 µP/cell to the cell cultures and incubated for further 24 h in standard culture conditions.

### 2.4. Cell Cultures in Fluidic Conditions

For these experiments, only cocultures were performed. Before seeding, channel slides (µ-Slides I 0.8 mm ibiTreat, ibidi) were coated with bovine collagen type I (Advanced Biomatrix, San Diego, CA, USA) to enhance cell adhesion. Then, 1.5 × 10^5^ cells of each cell line (D492 and D492HER2) were seeded in H14 medium containing 2% penicillin/streptomycin (Biowest, Nuaillé, France) and incubated in standard culture conditions. After 24 h, the slides were connected to a microfluidic system consisting of a perfusion set (Perfusion Set Red ID 1.6 mm, ibidi) filled with fresh H14 medium containing the µPs (µP-secAb or µP-antiH) and a Fluidic Unit connected to an ibidi Pump, controlled by a pump-control software (ibidi). The microfluidic system was kept at 37 °C and 5% CO_2_. The cultures were maintained for 24 h under a unidirectional flow rate fixed at 4.32 mL/min with a shear stress of 1.50 dyn/cm^2^ and a pressure of 7.9 mbar, as recommended by the manufacturer.

### 2.5. Evaluation of Microparticles Internalization

After being cultured in either static or fluidic conditions, the cells were washed three times with phosphate buffer saline (PBS) at room temperature (RT), fixed for 15 min with 4% paraformaldehyde (Sigma-Aldrich, St Louis, MO, USA) in PBS, and washed again with PBS (three times). Next, the fixed cells were permeabilized with 0.1% Triton X-100 (Sigma-Aldrich) in PBS for 10 min at RT, washed with PBS (three times), and blocked with 3% bovine serum albumin (BSA) (Sigma-Aldrich) in PBS for 40 min. PBS with 3% BSA was also employed to dilute the antibodies used in this work.

The cells from the monocultures were incubated with Alexa Fluor^®^ 546 Phalloidin (1:40, Life Technologies) to label actin microfilaments and, in the case of samples with µP-antiH, also with goat anti-mouse IgG1 secondary antibody Alexa Fluor^®^ 647 conjugate (1:150. Life Technologies) for 1 h at RT to detect µP-antiH.

To distinguish between D492 and D492HER2 in cocultures, the cells were first incubated with rabbit anti-HER2 monoclonal antibody (1:200, Cell Signaling, Danvers, MA, USA) overnight at 4 °C. Then, the samples were washed three times with PBS and incubated for 2.5 h at RT with Alexa Fluor^®^ 546 Phalloidin to label actin microfilaments to visualize the cell limit and chicken anti-rabbit IgG (H+L) Alexa Fluor^®^ 405 conjugate secondary antibody (1:150, Life Technologies) to label HER2 in the plasma membrane; for cells incubated with µP-antiH, goat anti-mouse IgG1 secondary antibody Alexa Fluor^®^ 647 conjugate was also used.

Finally, the cells were washed three times with PBS and maintained at 4 °C in PBS until evaluation under a confocal laser scanning microscope (CLSM. Olympus, Tokyo, Japan). Orthogonal projections of z-stacks of at least 100 cells for each cell line were evaluated in each replicate. The xyz sequentially acquired images allowed for assessment of whether the particles were inside the cells or attached to their surfaces.

### 2.6. Statistical Analyses

At least three replicates of each experiment were performed. To compare µP internalization in each experimental condition and for each cell line, ANOVA with post-hoc Tukey HSD test was used; *p* < 0.05 was considered statistically significant.

## 3. Results

### 3.1. Microparticles Characterization after Biofunctionalization

Biofunctionalization did not affect the size of µPs, as can be observed in TEM images (Figure 1A). Moreover, biofunctionalization was confirmed in two ways, microscopically and by analyzing the ζ-potential. As expected, under fluorescence microscopy, µP-secAb emitted far-red fluorescence, and µP-antiH emitted green fluorescence after incubation with an Alexa^®^ 488-conjugated secondary antibody (Figure 1B). Biofunctionalization was also confirmed by changes in the µP surface charge. Non-biofunctionalized polystyrene carboxylate µPs (µP-COOH) showed a ζ-potential value of −32.3 mV, whereas µP-secAb and µP-antiH increased their ζ-potential to smaller negative values of −11.23 mV and −11.5 mV, respectively (Figure 1C).

### 3.2. Microparticles Internalization by Cells

The internalization of µPs by cells was evaluated through the orthogonal images captured by a CLSM in both static and fluidic culture conditions (examples in Figure 2 and Figure 3, respectively). Staining of actin filaments was useful to visualize the cell perimeter and, together with the orthogonal projections, allowed us to clearly distinguish between internalized and non-internalized µPs. From these images, the number of cells with at least one internalized µP was scored.

As can be seen in Figure 4, in all conditions, the percentage of D492 cells with internalized microparticles was always higher than that of D492HER2 cells, indicating that D492 cells have an inherent superior capacity to internalize microparticles. Regarding the importance of specific biofunctionalization in µPs recognition and intake by the cells, the internalization related to non-specific binding due to the intrinsic cell endocytic capacity, was represented by the percentage of cells with internalized µPs biofunctionalized with the non-specific antibody (µPs-secAb). In contrast, the internalization related to the specific recognition of µPs by the cells was represented by the increase in the percentage of cells with internalized µPs when these were specifically functionalized (µP-antiH) to recognize a cell membrane receptor (HER2). For both cell lines, the biofunctionalization with a specific targeting antibody (antiH) resulted in higher internalization percentages than the biofunctionalization with a non-specific targeting antibody (secAb). The differences between static monoculture and coculture conditions were not significant. As expected, fluidic culture conditions globally decreased internalization, but again internalization was higher for microparticles which were specifically biofunctionalized (µP-antiH) than for those that were biofunctionalized with a non-specific antibody (µP-secAb). The increase in the percentage of cells with internalized µP-antiH was also higher for cells with HER2 overexpression (D492HER2) than for cells without HER2 overexpression (D492) (Figure 4). Remarkably, the increase in the percentage of cells with internalized µP-antiH with regard to µP-secAb internalization was higher in fluidic conditions than in static conditions, especially in D492HER2 cells (164% versus 77–99% in D492HER2 cells, and 100% versus 22–35% in D492 cells).

## 4. Discussion

Polystyrene µPs were successfully biofunctionalized with an anti-HER2 antibody or with a secondary antibody, as demonstrated by the detection of fluorescence under a microscope and by changes in their ζ-potential. The reduction in µPs electronegativity could help in their interaction with the plasma membrane, which contains negatively charged saccharides. In fact, it has been described that positively or slightly negatively charged particle surfaces favor cell intake [14,33,34,35].

The cell lines used in this study are isogenic breast epithelial cell lines. D492 was established by isolating suprabasal cells from a reduction mammoplasty and subsequently immortalizing them using the *E6* and *E7* oncogenes from the human papilloma virus [30,31]. D492 has stem cell properties evaluated by the ability to generate luminal and myoepithelial cells and, in 3D culture, to form branching ductal–alveolar-like structures. D492HER2 was generated by overexpressing the HER2 oncogene in D492 [32]. Interestingly, D492 showed higher percentages of µPs internalization than D492HER2. A possible explanation could be the differences in phenotype between these cells. D492HER2 cells undergo an epithelial to mesenchymal transition (EMT) and express mesenchymal markers, which could reduce their ability to internalize µPs [31,32]. Higher rates of µP internalization have also been described in MCF10A breast epithelial normal cells than in SKBR3 breast epithelial tumoral cells, which also overexpress HER2 [14].

The molecules used to bio-functionalize the µPs also influenced the internalization rates. For both cell lines, the percentage of cells with internalized µP-antiH was higher than that of cells with internalized µP-secAb. This result was expected, since we hypothesized that a specific recognition between the cells and the µPs, such as that afforded by antiH biofunctionalization, could result in an increase in µP internalization. Also as expected, the increase in the percentages of µPs internalization obtained with antiH biofunctionalization was higher for D492HER2 cells, which overexpress HER2, than for D492 cells, which express HER2 at much lower levels [32]. These results clearly indicate that when µPs are biofunctionalized with molecules that specifically interact with plasma membrane receptors, their internalization can be improved, and the level of improvement is related to the number of receptors that are present on the cell surface.

Finally, in relation to the culture conditions, our results indicate that the percentage of cells with internalized µPs clearly decreased in fluidic conditions compared with static conditions. This was probably because the establishment of prior transient contacts between the plasma membrane and the µPs may be more difficult in fluidic conditions or in high shear stress conditions [36,37]. One can hypothesize that in the case of dynamic cultures, specific targeting may be helpful to improve the internalization rates, as it may facilitate the frequency and strength of µPs-cell contacts. Our results agree with this hypothesis because, for both cell lines, the increase of µPs-antiH internalization was higher in fluidic than in static cocultures when compared with µPs-secAb internalization, which opens the door to scale-up studies.

The present results, together with previous research, emphasize the relevance of cell type, specific targeting, and culture conditions such as mono- or cocultures and static or fluidic, to the potential application of NPs and µPs for drug delivery to cancer cells.

## 5. Conclusions

The present study indicates that, for all culture conditions, µPs biofunctionalized with anti-HER2 antibodies were more efficiently internalized by cells expressing HER2 than µPs biofunctionalized with a non-specific antibody. Moreover, in fluidic culture conditions (simulating the blood stream), the specificity of targeting was still more useful for µP internalization, because the increase in internalization was higher for cells overexpressing HER2. Overall, the results presented emphasize the importance of targeting not only for directing µPs to the appropriate cells but also for achieving reasonable internalization rates, thereby opening the door to the use of microparticles as carriers for drug delivery.

## Figures and Tables

**Figure 1 pharmaceutics-11-00177-f001:**
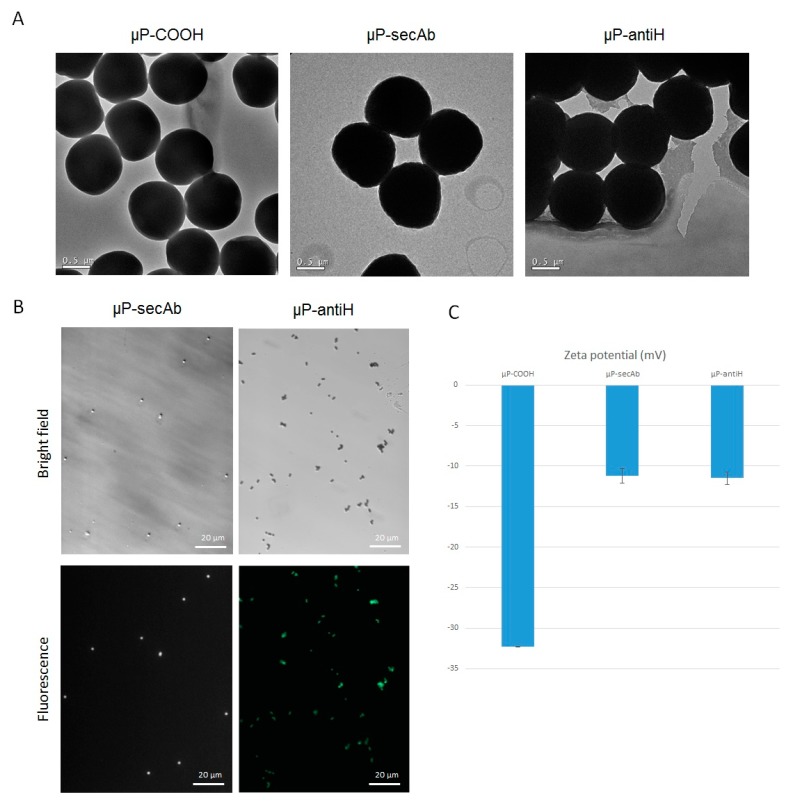
Characterization of microparticles (µP) biofunctionalization. (**A**) Transmission electronic microscopy (TEM) images of microparticles before (COOH) and after biofunctionalization (µP-secAb and µP-antiH). (**B**) Images of microparticles biofunctionalized with a secondary antibody (µP-secAb) or an anti-HER2 antibody (µP-antiH) in bright-field (upper panels) and fluorescence (lower panels) microscopy. (**C**) Zeta potential before (COOH) and after biofunctionalization (µP-secAb and µP-antiH).

**Figure 2 pharmaceutics-11-00177-f002:**
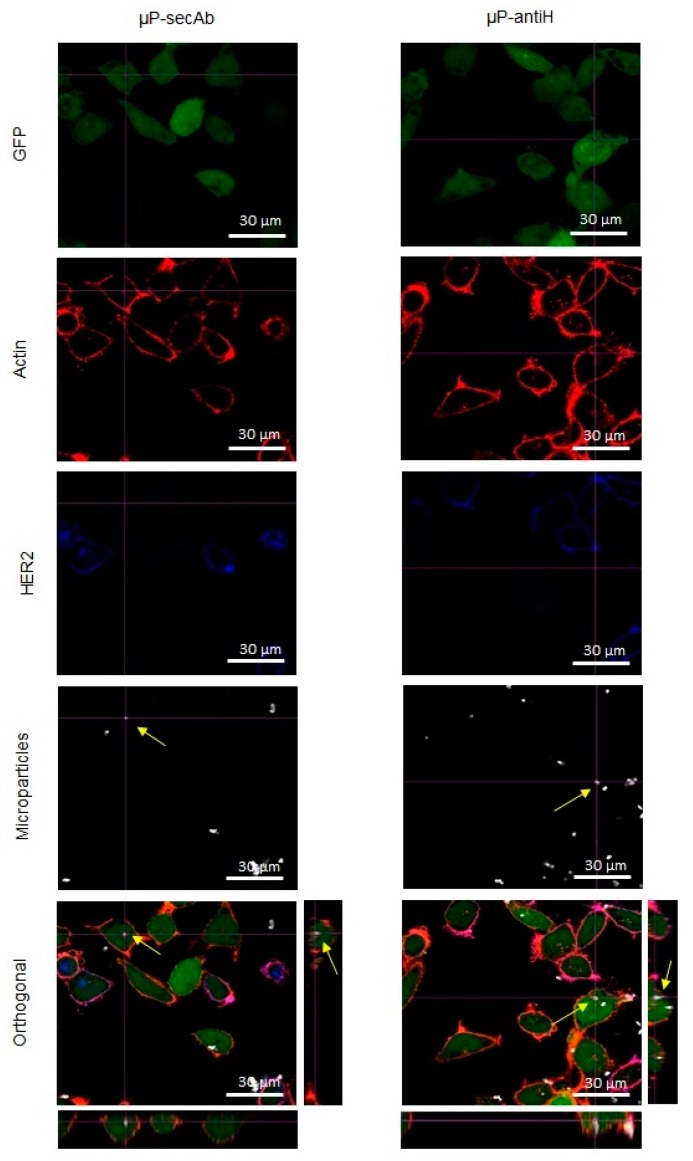
Immunofluorescence analysis by confocal laser scanning microscope (CLSM) of cells cultured in static conditions. Confocal images of D492 and D492HER2 cells cocultured in static conditions and incubated with microparticles biofunctionalized with a non-specific secondary antibody (µP-secAb) or a specific anti-HER2 antibody (µP-antiH). Cells, constitutively expressing green fluorescent protein (GFP, green), were incubated with Alexa Fluor^®^ 546 Phalloidin (red) to label actin microfilaments and Alexa Fluor^®^ 405 conjugate secondary antibody (blue) to label HER2 in the plasma membrane. The arrows point to some examples of µPs located inside the cells.

**Figure 3 pharmaceutics-11-00177-f003:**
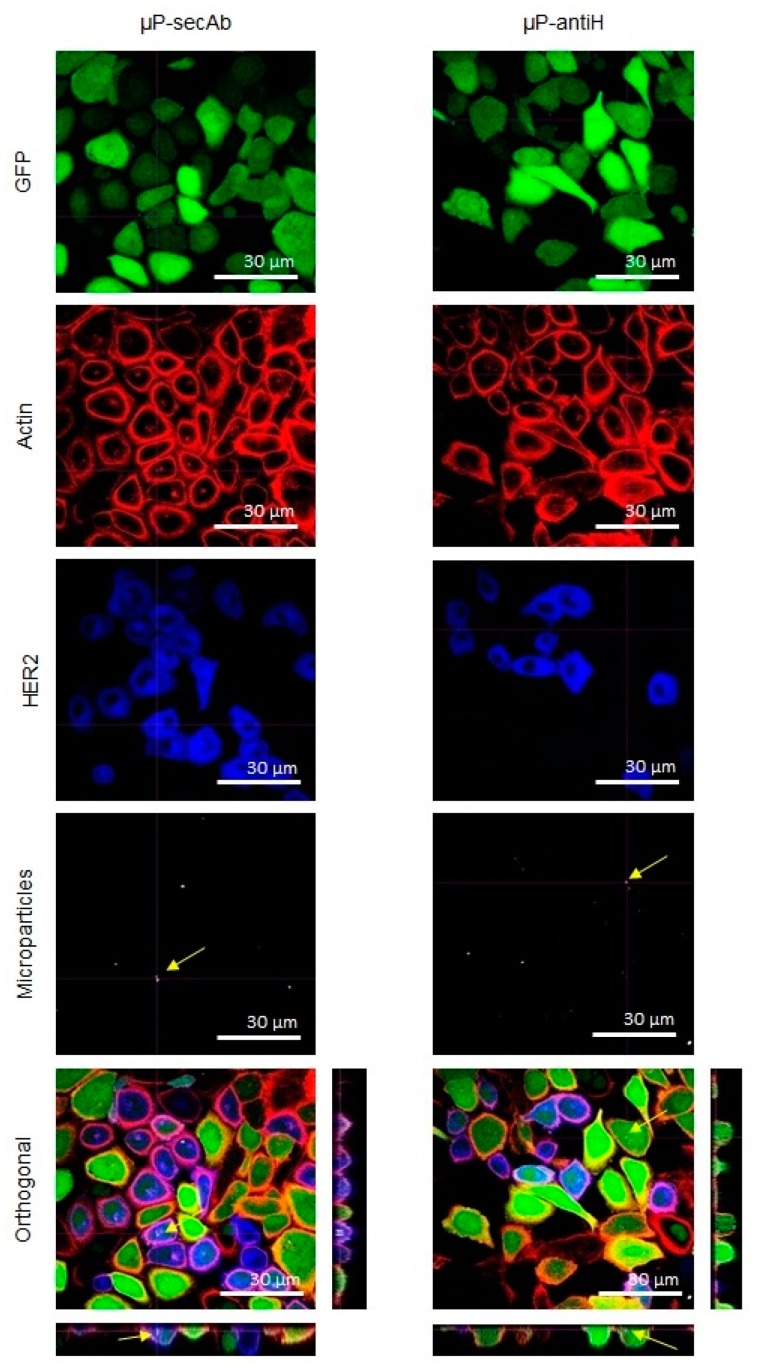
Immunofluorescence analysis by CLSM of cells cultured in fluidic conditions. Confocal images of D492 and D492HER2 cells cocultured in fluidic conditions and incubated with microparticles biofunctionalized with a non-specific secondary antibody (µP-secAb) or a specific anti-HER2 antibody (µP-antiH). The cells, constitutively expressing GFP (green), were incubated with Alexa Fluor^®^ 546 Phalloidin (red) to label actin microfilaments and Alexa Fluor^®^ 405 conjugate secondary antibody (blue) to label HER2 in the plasma membrane. The arrows point to some examples of µPs located inside the cells.

**Figure 4 pharmaceutics-11-00177-f004:**
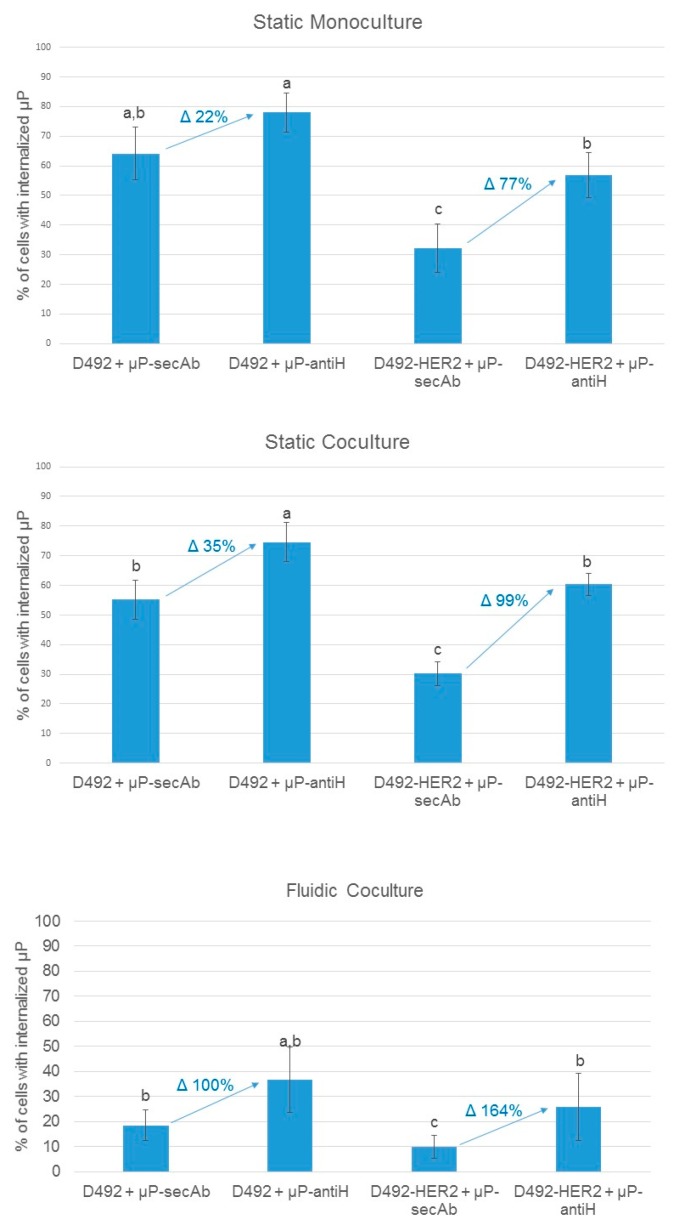
Microparticles internalization by monocultured or cocultured D492 and D492HER2 cells in static and fluidic conditions. Percentages of cells with internalized microparticles biofunctionalized with a non-specific secondary antibody (µP-secAb) or a specific anti-HER2 antibody (µP-antiH). Statistically significant differences are indicated with different letters on top of the bars. An increase of µPs internalization, as a percentage, is indicated in blue together with blue arrows.

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
