# Peer review of "Cell Internalization in Fluidic Culture Conditions Is Improved When Microparticles Are Specifically Targeted to the Human Epidermal Growth Factor Receptor 2 (HER2)"

_pharmaceutics, 2019, doi:10.3390/pharmaceutics11040177_

Reviewer 1 Report

1. The title is generic for all microparticles, but the authors focus only in specific (one) size. Therefore, they will need to perform more studies using at least couple more particle sizes (maybe 1.5 and 2 um). Also, will be good to have a direct comparison with nanoparticles. Moreover, at least one more biofunctionalisation molecule needs to be studied, otherwise the authors can’t state that the method is working for "everything".

2. The microparticles (before and after functionalisation) needs extra characterisation studies, such as: FTIR, DSC, TEM, particle sizing, and stability studies over time and under different temperatures.

3. Comment the possibility to scale up this method.

4. Up-to-date references are needed in the introduction and discussion.

Author Response

Responses to Reviewer 1

1. The title is generic for all microparticles, but the authors focus only in specific (one) size. Therefore, they will need to perform more studies using at least couple more particle sizes (maybe 1.5 and 2 um). Also, will be good to have a direct comparison with nanoparticles. Moreover, at least one more biofunctionalisation molecule needs to be studied, otherwise the authors can’t state that the method is working for "everything".

RESPONSE: According to the reviewer we have changed the title to: “Polystyrene 1 um microparticles biofunctionalization with targeting molecules improves their internalization by cells in fluidic culture conditions”.

As can be seen in the figure enclosed, we performed the same experiments with 3 um particles. We decided not to include these results because for fluidic conditions internalization was too low. However, if the reviewer and the Academic Editor consider that 3 um results should be included in the main text or in the supplementary material we can do it.

The aim of the present work was not compare NP vs uP, but to compare static vs dynamic cultures. We agree with the referee that it would be interesting to compare the behaviour of different size of NPs and uPs, but this is out of the objective of the present study.

2. The microparticles (before and after functionalisation) needs extra characterisation studies, such as: FTIR, DSC, TEM, particle sizing, and stability studies over time and under different temperatures.

RESPONSE: We agree with the reviewer that microparticles characterization is a key point in this kind of studies. Antibody recognition is a useful and widely used method to detect the presence of specific proteins, and for this reason we have followed the same protocol performed in previous studies with these particles, functionalized using a specific commercial kit. As can be seen in figure 1, biofunctionalized microparticles showed fluorescence emission, only possible if they were functionalized by a molecule recognized by the antibody. Moreover, the reduction in electronegativity, detected through the analysis of the ζ-potential, in biofunctionalyzed microparticles, is also indicative of a successful functionalization. Regarding particle sizing, the coefficient of variance is 3% for the microparticles used in the present study (data from the supplier, Polysciences), which means a high homogeneity, higher than for the majority of nanoparticles. We think that changes in particle size as a result of biofunctionalization are more substantial in the case of nanoparticles but have a low impact in the case of microparticles.

3. Comment the possibility to scale up this method.

RESPONSE: We have included a change (in italics) in the sentence at the end of the Discussion 4th paragraph. “Our results agree with this idea because, for both cell lines, the increase of µPs-antiH internalization was higher in fluidic than in static cocultures when compared with µPs-secAb, which opens the door to scale up studies.”

4. Up-to-date references are needed in the introduction and discussion.

RESPONSE: We have included two new references (27 and 37) on fluidic conditions.

Reviewer 2 Report

Authors used anti-HER2 antibody as active targeting ligand to decorate the macroparticles (carboxylate polystyrene) and proposed to increase the internalization by HER2 positive cancer cells. The HER2 antibody was well established and verified in clinic for both active cancer cell targeting and antitumor efficacy, which laid the ground for this study, however, the data provided in this manuscript were too premature (or lack of strong evidence) to get the solid conclusion, in addition, the methodological approaches and data presentation were merely enough to achieve the publication requirements.

Some comments were exampled below for authors’ reference.

Major concerns:

1: The pharmaceutical of antibody-modified macroparticle is lack of good characterizations, such as mean particles size, antibody coupling percentage, and stability, etc.

2: The images resolution were poor in the whole manuscript.

3: No quantitative analysis method was used in this manuscript to show the difference between groups, like flow cytometry.

4: The biocompatibility test should be done for antibody-modified macroparticle.

5: The difference in both Figure 2 and 3 are not sufficient to get the conclusion shown in Figure 4.

6: How did authors get the data in Figure 4, and what methods did authors use to generate the data?

Minor comments:

The source of D492 and D492HER2 should be provided in the manuscript.

How did author calculate the “μPs were added at a proportion of 45 μP/cell” in the line 96.

The cell culture medium condition should be checked as serum free or not.

Author Response

Responses to Reviewer 2

Authors used anti-HER2 antibody as active targeting ligand to decorate the macroparticles (carboxylate polystyrene) and proposed to increase the internalization by HER2 positive cancer cells. The HER2 antibody was well established and verified in clinic for both active cancer cell targeting and antitumor efficacy, which laid the ground for this study, however, the data provided in this manuscript were too premature (or lack of strong evidence) to get the solid conclusion, in addition, the methodological approaches and data presentation were merely enough to achieve the publication requirements.

Some comments were exampled below for authors’ reference.

Major concerns:

1: The pharmaceutical of antibody-modified macroparticle is lack of good characterizations, such as mean particles size, antibody coupling percentage, and stability, etc.

RESPONSE: We agree with the reviewer that microparticles characterization is a key point in this kind of studies. Antibody recognition is a useful and widely used method to detect the presence of specific proteins, and for this reason we have followed the same protocol performed in previous studies with these particles, functionalized using a specific commercial kit. As can be seen in figure 1, biofunctionalized microparticles showed fluorescence emission, only possible if they were functionalized by a molecule recognized by the antibody. Moreover, the reduction in electronegativity, detected through the analysis of the ζ-potential, in biofunctionalyzed microparticles, is also indicative of a successful functionalization. Regarding particle sizing, the coefficient of variance is 3% for the microparticles used in the present study (data from the supplier, Polysciences), which means a high homogeneity, higher than for the majority of nanoparticles. We think that changes in particle size, due to biofunctionalization, are more substantial in the case of nanoparticles but have a low impact in the case of microparticles.

2: The images resolution were poor in the whole manuscript.

RESPONSE: We have received a formatted version of the manuscript and we believe that in this version the images have enough quality.

3: No quantitative analysis method was used in this manuscript to show the difference between groups, like flow cytometry.

RESPONSE: From our experience in internalization studies, we have observed that microparticles can be internalized by the cell but also adhered to the cell surface. For this reason, we performed the analysis of orthogonal projections of z-stacks under CLSM, which allowed us to clearly distinguish between internalized or surface-adhered microparticles. We analysed at least 100 cells for each experimental condition to obtain quantitative results. 

4: The biocompatibility test should be done for antibody-modified macroparticle.

 RESPONSE: The biocompatibility of functionalised and non-functionalised Polybead Carboxilate Microspheres Polyscience has been previously tested in our laboratory in previous studies in both cell lines and mouse embryos (Novo et al, DOI: 10.1007/s10544-013-9766-8; Patiño et al, DOI: 10.1038/srep11371 and DOI: 10.2147/IJN.S34635 and Mora-Espí https://doi.org/10.1038/s41598-018-35913-3).

5: The difference in both Figure 2 and 3 are not sufficient to get the conclusion shown in Figure 4.

RESPONSE: As indicated in the response to point 3, we analysed at least 100 cells for each experimental condition to obtain quantitative results. Figures 2 and 3 are only an example of the analysis performed in one of these cells.

6: How did authors get the data in Figure 4, and what methods did authors use to generate the data?

RESPONSE: Please see response to point 3.

Minor comments:

The source of D492 and D492HER2 should be provided in the manuscript.

RESPONSE: As stated in the manuscript, these cell lines were obtained and modified in the laboratory of one of the authors (T. Gudjonsson), as indicated by the references cited in the manuscript. For this reason no commercial source is indicated.

How did author calculate the “μPs were added at a proportion of 45 μP/cell” in the line 96.

RESPONSE:

The cell culture medium condition should be checked as serum free or not.

 RESPONSE: As indicated in the “Cell lines” section, cells were cultured in “serum free” H14 medium, as previously described (see references 30,31).

Round  2

Reviewer 1 Report

1. Title is long and confusing the reader. Therefore, will be nice the authors to choose an "attractive" title that represents their studies. 

2. As I mentioned in my previous comments, the microparticles (before and after functionalisation) needs extra characterisation studies, such as: FTIR, DSC, TEM, particle sizing, and stability studies over time and under different temperatures. The authors haven’t performed any of these studies. 

Author Response

2nd revision

Reviewer 1

Comment 1: Title is long and confusing the reader. Therefore, will be nice the authors to choose an "attractive" title that represents their studies. 

RESPONSE: We have changed the title: “Polystyrene 1 μm microparticles biofunctionalization with targeting molecules improves their internalization by cells in fluidic culture conditions” by this one:

“Cell internalization in fluidic culture conditions is improved when microparticles are specifically targeted to HER2”

Comment 2: As I mentioned in my previous comments, the microparticles (before and after functionalisation) needs extra characterisation studies, such as: FTIR, DSC, TEM, particle sizing, and stability studies over time and under different temperatures. The authors haven’t performed any of these studies.

RESPONSE: The particles used in the present study are commercial, and their characteristics are described by the supplier (http://www.polysciences.com/default/polybead-carboxylate-microspheres-100956m). Moreover, we used an optimized commercial kit from the same supplier to functionalize the microparticles, as in previous studies of our group (Novo et al, DOI: 10.1007/s10544-013-9766-8; Patiño et al, DOI: 10.1038/srep11371 and DOI: 10.2147/IJN.S34635 and Mora-Espí https://doi.org/10.1038/s41598-018-35913-3).

       Taking into account the reviewer suggestion, we have added the characterization of microparticles size by TEM. The changes introduced due to these new data can be seen in:

- Materials methods: The size of µPs before and after biofunctionalization was analyzed by transmission electronic microscopy, TEM (JEOL, JEM 2011).

- Results: Biofunctionalization did not affect the size of µPs, as can be observed in TEM images (Figure 1A).

- Figure 1. We have added TEM images of microparticles, before and after biofunctionalization.

Reviewer 2 Report

I do not free that authors fully and well answer those questions, especial for the critical question on the analytical method used for internalization studies, the data was not convincing enough just by the counting the microparticles instead of any more scientific methods. If the paper gets publication without further justification, some uncertain issues will be asked by the reader such like what are those bright dots in those figures without being highlighted by arrow? Are they some bound microparticles lack of washing? if so, how to differentiate the internalization coming from the antibody-directed or non-specific binding. In addition, one of the questions is not answer? "How did author calculate the “μPs were added at a proportion of 45 μP/cell” in the line 96."

Author Response

2nd revision

Reviewer 2

Comment 1: I do not free that authors fully and well answer those questions, especial for the critical question on the analytical method used for internalization studies, the data was not convincing enough just by the counting the microparticles instead of any more scientific methods.

RESPONSE: The analysis of orthogonal projections of CLSM images allows to obtain a precise quantification of the percentage of cells with internalized microparticles. This idea is based in previous studies of our group (Patiño et al, Int J Nanomedicine 2012:7 1-12. 10.2147/IJN.S34635), where we showed that flow cytometry and CSLM analysis give similar results of microparticles internalization (please, see in the attached file the figure 6 from this article). For this reason, we have used the CLSM analysis in other studies where the percentage of cells with internalized microparticles were quantified (Mora-Espí et al, Scientific Reports 2018:8: 17617. 10.1038/s41598-018-35913-3).

Comment 2: If the paper gets publication without further justification, some uncertain issues will be asked by the reader such like what are those bright dots in those figures without being highlighted by arrow? Are they some bound microparticles lack of washing?

RESPONSE: Bright dots in the figures are microparticles. Those highlighted by arrows are examples to show that they are surrounded by the plasma membrane, indicating that they are internalized, as it can be seen by the orthogonal projections.  We have not highlighted the other microparticles to avoid a figure full of arrows. To clarify what is observed in figures 2 and 3 are examples, we have modified the text in results section (point 3.2) and the figures legend, as follows:

Results (3.2):

The internalization of µPs by cells was evaluated through the orthogonal images captured on the CLSM in both static and fluidic culture conditions (See examples in Figures 2 and 3, respectively).

Figure legends:

Arrows point to some examples of µPs located inside the cells. Scale bar: 30 µm.

As the reviewer indicates, some particles can be on the surface of the cells instead internalized. This is why the orthogonal images of CLSM are useful, allowing to clearly distinguish between internalized and surface attached microparticles. We have modified the text in section 2.5 to clarify it.

To distinguish between D492 and D492HER2 in cocultures, cells were first incubated with rabbit anti-HER2 monoclonal antibody (1:200. Cell Signaling, Danvers, MA, USA) overnight at 4°C. Then, the samples were washed three times with PBS and incubated for 2.5 h at RT with Alexa Fluor® 546 Phalloidin to label actin microfilaments to visualize the cell limit, chicken anti-rabbit IgG (H+L) Alexa Fluor® 405 conjugate secondary antibody (1:150. Life Technologies) to label HER2 in the plasma membrane, and, only for cells incubated with µP-antiH, goat anti-mouse IgG1 secondary antibody Alexa Fluor® 647 conjugate.

Finally, cells were washed three times with PBS and maintained at 4°C in PBS until their evaluation under a confocal laser scanning microscope (CLSM. Olympus, Tokyo, Japan). Orthogonal projections of z-stacks of at least 100 cells of each cell line were evaluated in each replicate. The x-y-z sequential acquired images, allow to assess if the particles are inside the cells or attached to their surface.

if so, how to differentiate the internalization coming from the antibody-directed or non-specific binding.

The internalization related with non-specific binding (due to the intrinsic cell endocytic capacity) is represented by the percentage of cells with internalized microparticles biofunctionalized with the unspecific antibody. The antibody-targeted internalization is represented by the increase in the percentage of cells with internalized microparticles when these are specifically functionalized (anti HER2 antibody) to recognize a cell membrane receptor (HER2). This is represented in figure 4 as percentages of increase in microparticles internalization.

Comment 3: In addition, one of the questions is not answer? "How did author calculate the “μPs were added at a proportion of 45 μP/cell” in the line 96."

RESPONSE: The reviewer is right, we did not answer this question. To clarify this issue we have modified the text as follows:

Original text:

To analyze µP internalization, µPs (µP-antiH or µP-secAb) were sonicated for 5 min and then added to the cell cultures and incubated for further 24 h in standard culture conditions. In all the experiments, µPs were added at a proportion of 45 µP/cell.

Modified text:

To analyze µP internalization, µPs (µP-antiH or µP-secAb) were sonicated for 5 min, diluted (1:100), counted with a hemocytometer and then added, at a proportion of 45 µP/cell, to the cell cultures and incubated for further 24 h in standard culture conditions.

Round  3

Reviewer 2 Report

I agreed with authors' responses and was fine with detailed information (author included in this version now).

No more revision is needed from mine.

Author Response

To clarify the question "how to differentiate the internalization coming from the antibody-directed or non-specific binding", we have added the following paragraph at the manuscript:

Regarding the importance of specific biofunctionalization on µPs recognition and intake by the cells, the internalization related with non-specific binding, due to the intrinsic cell endocytic capacity, is represented by the percentage of cells with internalized µPs biofunctionalized with the non-specific antibody (µPs-secAb). In contrast, the internalization related with the specific recognition of µPs by the cells is represented by the increase in the percentage of cells with internalized µPs when these are specifically functionalized (µP-antiH) to recognize a cell membrane receptor (HER2).